# Antioxidant and Cytoprotective Capacities of Various Wheat (*Triticum aestivum* L.) Cultivars in Korea

**DOI:** 10.3390/foods11152338

**Published:** 2022-08-05

**Authors:** Huijin Heo, Hana Lee, Jinhee Park, Kyeong-Hoon Kim, Heon-Sang Jeong, Junsoo Lee

**Affiliations:** 1Department of Food Science and Biotechnology, Chungbuk National University, Cheongju 28644, Korea; 2Wheat Research Team, National Institute of Crop Science, Rural Development Administration, Wanju 55365, Korea

**Keywords:** wheat, antioxidant, cytoprotection, principal component analysis, hierarchical cluster

## Abstract

Whole wheat (*Triticum aestivum* L.) is a rich source of bioactive compounds with health benefits such as antioxidant, anti-inflammatory, and anti-proliferative activities. In this study, we evaluated the antioxidant and cytoprotective capacities of 41 wheat cultivars. The antioxidant capacities of whole wheat grain extracts (WWE) from various wheat cultivars were evaluated using total phenolic content and ABTS and DPPH radical-scavenging activities. The relative antioxidant capacity index (RACI) was calculated to integrate different chemical methods of antioxidant capacity. The cytoprotective capacity of the WWE was investigated using hepatocytes (HepG2), myoblasts (C2C12), and endothelial (EA.hy926) cells. The RACI was the highest and lowest for Dajoong (1.71) and Topdong (−1.96), respectively. Pearson correlation tests were conducted to determine the correlation between the antioxidant and cytoprotective activities. However, no significant correlations between RACI and the cytoprotective capacities were ascertained. Principal component analysis revealed that the first two components represented 68.93% of the total variability. Hierarchical cluster analysis classified WWE into three groups based on measured parameters. The results of this study highlight the variation in the antioxidant and cytoprotective capacities of 41 different wheat cultivars. This study provides basic information that can guide decisions in wheat breeding programs for the development of functional food crops and new dietary ingredients.

## 1. Introduction

Wheat (*Triticum aestivum* L.) is the second most widely consumed cereal after rice. Wheat contains various bioactive compounds, including vitamins, phenolic compounds, phytosterols, carotenoids, and minerals [1,2]. Studies have indicated that the intake of whole wheat may be associated with beneficial health effects such as reduced risk of coronary heart diseases, cancers, and type 2 diabetes [3,4]. Antioxidants inhibit the oxidation of other molecules by inhibiting the initiation or propagation steps of oxidation chain reactions [5]. Whole wheat and its products are rich in natural antioxidants and can be used for the development of functional foods. Many studies have demonstrated the antioxidant activities and phytochemical content of whole wheat. Adom et al. (2003) reported that the ferulic acid and carotenoid contents among 11 diverse varieties of wheat were statistically different [2]. Okarter et al. (2010) reported that the antioxidant activities and phytochemical contents of six varieties of whole wheat were significantly different [6]. Ebeid et al. (2015) investigated the hepatoprotective effects of wheat extract against carbon tetrachloride (CCl_4_)-induced oxidative stress [7]. Whent et al. (2012) demonstrated that whole wheat flour from five wheat cultivars exerted anti-inflammatory and anti-proliferative activities in HT-29 human colon cancer cells. [8]. However, the relationship between the antioxidant properties and cytoprotective activities of various whole wheat cultivars has not been fully investigated.

The objective of this study is to determine the antioxidant and cytoprotective capacities of various wheat cultivars bred in Korea. This study also aims to analyze the correlation between antioxidant and cytoprotective activities. The results of this study provide useful information for the development of wheat varieties as functional food materials for improving human health.

## 2. Materials and Methods

### 2.1. Plant Materials

Whole wheat cultivars were provided by the Rural Development Administration of the Republic of Korea in 2019. Korean wheat cultivars were sown in randomized complete blocks with 3 replicated in the upland crop experimental farm of the National Institute of Crop Science (NICS) of the Rural Development Administration (RDA, Republic of Korea). The seeds were sown on 25 October 2018 and plots were combine-harvested on 16 June 2019. Fertilizer was applied at 9.1:7.4:3.9 kg/10a (N:P:K) before sowing. Weeds, insects, and disease were stringently controlled. No supplemental irrigation was applied. Grain from each plot was dried using forced air driers and bulked from replications to provide grain for quality analysis. A list of 41 different wheat cultivars and their cross-combination information is shown in Table 1.

### 2.2. Preparation of Whole Wheat Grain Extracts (WWE)

Whole wheat grain (30 g) from each cultivar was milled using a Wiswell grinder (SP-7426, Supreme Electric Manufacture Co., Ltd., Guangzhou, China) for 30 s. Whole wheat powder (7 g) was extracted using 150 mL methanol for 16 h at room temperature using a shaker. The methanol extracts were evaporated under vacuum, and the concentrates were dissolved in dimethyl sulfoxide (DMSO) to obtain a concentration of 100 mg/mL. The WWE was filtered through a 0.22 μm sterile filter and stored at −20 °C until further use.

### 2.3. Antioxidant Capacities and Total Phenolic Content in WWE

The total phenolic content (TPC) and free-radical-scavenging activities (DPPH and ABTS assays) of the WWE were measured as described by Lissi et al. and Zhang et al. [9,10]. TPC was expressed as mg gallic acid equivalent and DPPH and ABTS radical-scavenging activities were expressed as Trolox equivalent antioxidant capacity. The relative antioxidant capacity index (RACI) was calculated to integrate different chemical methods of antioxidant capacity and TPC in WWE [11]. The RACI was calculated using the following equation:Standard score of each experiment = (raw data − mean)/standard deviation
RACI = Average of standard score(1)


### 2.4. Cell Culture

Hepatocytes (HepG2), myoblasts (C2C12), and endothelial cells (EA.hy926) were purchased from American Type Culture Collection (Rockville, MD, USA). The cells were grown in Dulbecco’s modified Eagle medium containing 100 U/mL penicillin, 100 μg/mL streptomycin, and 10% heat-inactivated fetal bovine serum in 5% CO_2_ humidified air at 37 °C.

### 2.5. Cytoprotective Activities in Hepatocytes

Human hepatocytes (HepG2) were seeded in 96-well plates (1 × 10^5^ cells/well) for 16 h. The culture medium was changed with or without WWE and incubated for 24 h. The cells were then treated with 500 μM *tert*-butyl hydrogen peroxide (*t*-BHP) for 3 h. To determine cell viability, 3-(4,5-dimethylthiazole-2-yl)-2,5-diphenyl-tetrazolium bromide (MTT) reagent was added to each well. After 2 h, the formazan product was dissolved in DMSO. Absorbance was measured at 550 nm by using a spectrophotometer.

### 2.6. Cytoprotective Activities in Myoblasts

Mouse myoblasts (C2C12) were seeded in 96-well plates (1.6 × 10^4^ cells/well) for 16 h. The culture medium was changed with or without WWE and incubated for 2 h. The cells were then treated with 600 μM hydrogen peroxide (H_2_O_2_) with WWE for 24 h. For determining cell viability, MTT reagent was added to each well for 2 h, and the formazan product was dissolved in DMSO. Absorbance was measured at 550 nm using a spectrophotometer.

### 2.7. Cytoprotective Activities in Endothelial Cells

Human endothelial cells (EA.hy926) were seeded in 96-well plates (1.4 × 10^4^ cells/well) for 24 h. The culture medium was changed with or without WWE and incubated for 2 h. The cells were then treated with 600 μM H_2_O_2_ with WWE for 24 h. To determine cell viability, MTT reagent was added to each well for 2 h, and the formazan product was dissolved in DMSO. Absorbance was measured at 550 nm using a spectrophotometer.

### 2.8. Calculation of Cytoprotective Activities

The percentage of cell viability was calculated using the following equation:Cell viability (% of control) = 100 − {(*A*_control_ − *A*_sample_)/*A*_control_ × 100}(2)
where *A* is absorbance. The cytoprotective activities were calculated to calibrate the viability of each 96-well plate. The calculation formula is as follows:Cytoprotective activities (%) = (*CV*_sample_ − *CV*_nc_)/(*CV*_control_ – *CV*_nc_) × 100(3)
where *CV* is the cell viability (% of control) and nc is the negative control. When the cytoprotective activity was less than 0, it was indicated as 0.

### 2.9. Statistical Analysis

Pearson correlation analysis was performed using SPSS version 18 (SPSS Inc., Chicago, IL, USA). Principal component analysis (PCA) was conducted using PAST 4.03 software [12]. Hierarchical clustering analysis was performed using R studio statistical software with a heat map visualization.

## 3. Results and Discussion

### 3.1. Antioxidant and Cytoprotective Capacities

Various solvents (ethanol, methanol, acetone, hexane, chloroform, ethyl acetate, and water) were tested to select the most effective solvent for extracting antioxidants from whole wheat (Appendix A). Methanol was the most effective solvent, followed by ethanol (Appendix A). According to Lopez-Perea et al. (2019), methanolic extracts of wheat bran had the highest activities in DPPH and ABTS assays [13]. Descriptive statistics of antioxidant properties and cytoprotective activities in the 41 wheat cultivars are shown in Table 2, and raw data are added in Appendix A. The cultivar with the highest TPC was Dajoong (12.23 gallic acid equivalents (GAE) mg/g residue) and the lowest TPC was Ol (9.62 GAE mg/g residue). The DPPH radical-scavenging activities of the 41 wheat cultivars ranged from 3.17 (Gobun) to 3.89 (Milsung) Trolox equivalents (TE) mg/g residue. The ABTS radical-scavenging activities of the WWE ranged from 27.73 (Topdong) to 57.16 (Dajoong) TE mg/g residue. Because of the different in vitro reactions of the antioxidant activities for determining WWE antioxidant properties, the values may vary substantially. Therefore, for observing the trend and ranking the antioxidant properties of the WWE, the RACI was calculated using Equation (1). The RACI (Figure 1) was the highest for Dajoong (1.71), followed by Dabun (1.04) and Eunpa (0.80). Topdong (−1.96) and Ol (−1.38) had the lowest indexes among the cultivars. Several studies have demonstrated that wheat has different antioxidant capacities depending on its variety [2,6,14]. According to Abdel-Aal et al. (2001), ferulic acid content varies significantly in wheat cultivars grown in different environments [15]. Organic or conventional growth conditions affect phenolic compounds in wheat cultivars [14].

The cytoprotective effect of WWE was observed on different cell lines, such as hepatocytes (HepG2), myoblasts (C2C12), and endothelial cells (EA.hy926). WWE had the most prominent cytoprotective effect on the hepatocytes. Early studies have shown that whole wheat can be used as a nutraceutical agent for the treatment of hepatotoxicity [7,16].

### 3.2. Correlations among Antioxidant Properties and Cytoprotective Activities

This study demonstrated the antioxidant and cytoprotective capacities of WWE from 41 different wheat cultivars by using TPC, DPPH, ABTS, and MTT assays. Pearson correlation tests were conducted to determine the correlation between the antioxidant and cytoprotective activities of WWE (Table 3). However, no significant correlations between RACI and cytoprotective activities were ascertained. This might be due to the presence of other bioactive compounds including alkylresorcinols, phytosterols, and policosanols in wheat. Phytosterols are structurally related to cholesterol and reduce serum low-density lipoprotein cholesterol levels [17]. According to a previous report, β-sitosterol inhibited muscle atrophy in muscle atrophy C2C12 cells [18]. Alkylresorcinols exhibit antifungal and antibacterial activities [19,20]. Octacosanol enhances the migration and proliferation of human umbilical vein endothelial cells [21]. However, De Jong et al. (2008) found that phytosterol consumption did not affect antioxidative enzyme activity, endothelial dysfunction, and low-grade inflammation [22]. Therefore, further research is necessary to study the relationship between cytoprotective activity and phytochemical content. 

### 3.3. Principal Component Analysis (PCA)

PCA was used to define the discriminant effects of RACI and cytoprotective activities between the wheat cultivars. PCA is a technique that allows the visualization of the information present in the original data as much as possible by reducing data dimensionality. The data were subjected to PCA and the results show the most significant PCs (Table 4). Of the four principal components (PCs), PC1 and PC2 had eigenvalues > 1 and could explain 68.93% of the total cumulative variability (only eigenvalues of >1 are considered significant descriptors of data variance). PC3 and PC4 yielded progressively smaller eigenvalues (0.83 and 0.41, respectively) and did not explain significant variability in the data. PC1 describes 36.89% of the variance in the data set and indicates that it has high contributions from the Myoblasts and Endothelial cells variables. PC2 accounted for 32.04% of total variance, which was strongly influenced by the RACI and Hepatocytes variables. Specific patterns of correlation between the variables studied can be visualized when one compares plots between PCs [23]. The first two PCs, which contributed 68.93% of the total variance, were plotted as biplots on PC1 and PC2 on the X- and Y-axes, respectively, to detect the association between variables (Figure 2). It was clear that the RACI and Hepatocytes variables were grouped together on the on the right upper side of the biplot, suggesting that these two variables had a positive correlation. The negative correlation between the Endothelial cells and Myoblasts variables was revealed. Taken together, PCA results showed a wide range of variations with regard to antioxidant and cytoprotective capacities even though Myoblasts and Endothelial cells in PC1 were found to be the most discriminant variables to classify cultivars. Therefore, we then used hierarchical clustering analysis to classify cultivars depending on bioactivities.

### 3.4. Clustering Analysis

Hierarchical clustering analysis (HCA) is a multivariate analytical tool used to cluster samples based on dissimilar characteristics and is displayed as a dendrogram. The 41 wheat cultivars were classified into three clusters according to their antioxidant and cytoprotective activities (Figure 3 and Table 5). Group A contained 13 cultivars and had the highest RACI and cytoprotective activities in hepatocytes and myoblast. Group B contained 25 cultivars and had the highest cytoprotective activities in the endothelial cells. Group C contained three cultivars and had the relatively lowest antioxidant and cytoprotective activities in the hepatocytes and endothelial cells. 

## 4. Conclusions

In this study, the antioxidant and cytoprotective capacities of 41 different wheat cultivars were determined. Dajoong (1.71) had the highest and Topdong (−1.96) had the lowest RACI among the 41 wheat cultivars. Statistical analysis revealed a significant positive correlation between RACI and antioxidant activity. However, no significant correlations between the antioxidant and cytoprotective capacities were ascertained. Therefore, further investigations are warranted to gain more insights into the relationships between phytochemical content and cytoprotective activities. The correlation between RACI and cytoprotective activities was weak, which could be investigated by further studying the relationships between phytochemical content and cytoprotective activities. From the PCA results, the most discriminant variables were Myoblasts and Endothelial cells in PC1 and the RACI and Hepatocytes variables in PC2. HCA clustered the 41 wheat cultivars into three classifications. Hierarchical clustering analysis indicated that the 41 wheat cultivars could be clustered into three classifications. This study provides supportive information on the relationship between the antioxidant and cytoprotective capacities of WWE. In addition, this result provides a detailed examination of cultivar effects on potential antioxidant and cytoprotective capacities in wheat, and the results can be used for screening and breeding purposes. Further studies are in progress to improve the understanding of the relationship between bioactivities and phytochemical contents.

## Figures and Tables

**Figure 1 foods-11-02338-f001:**
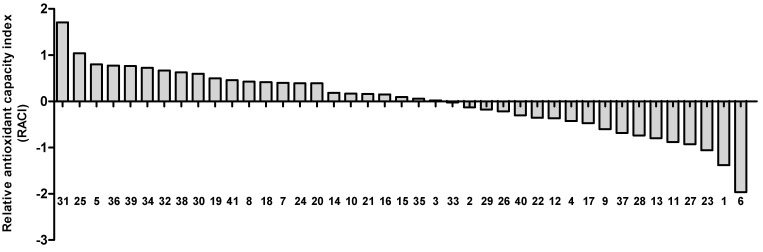
Relative antioxidant capacity index (RACI) of whole wheat extracts (WWE).

**Figure 2 foods-11-02338-f002:**
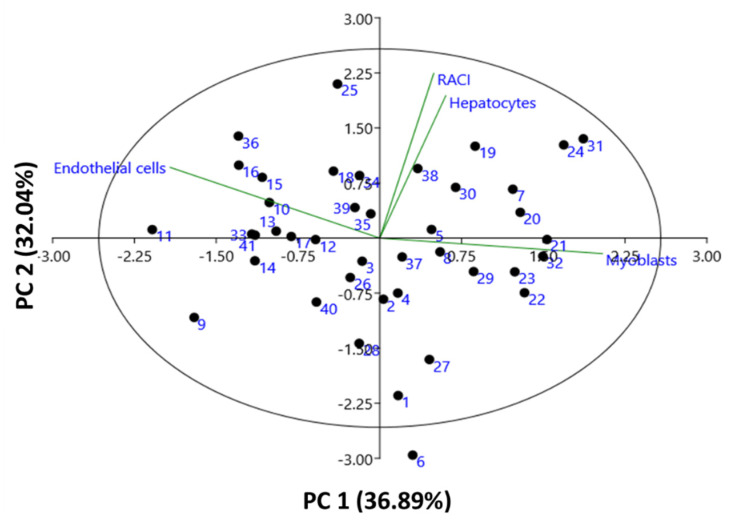
2D scatter diagram of principal component analysis (PCA) of whole wheat extracts (WWE) based on antioxidant and cytoprotective capacities.

**Figure 3 foods-11-02338-f003:**
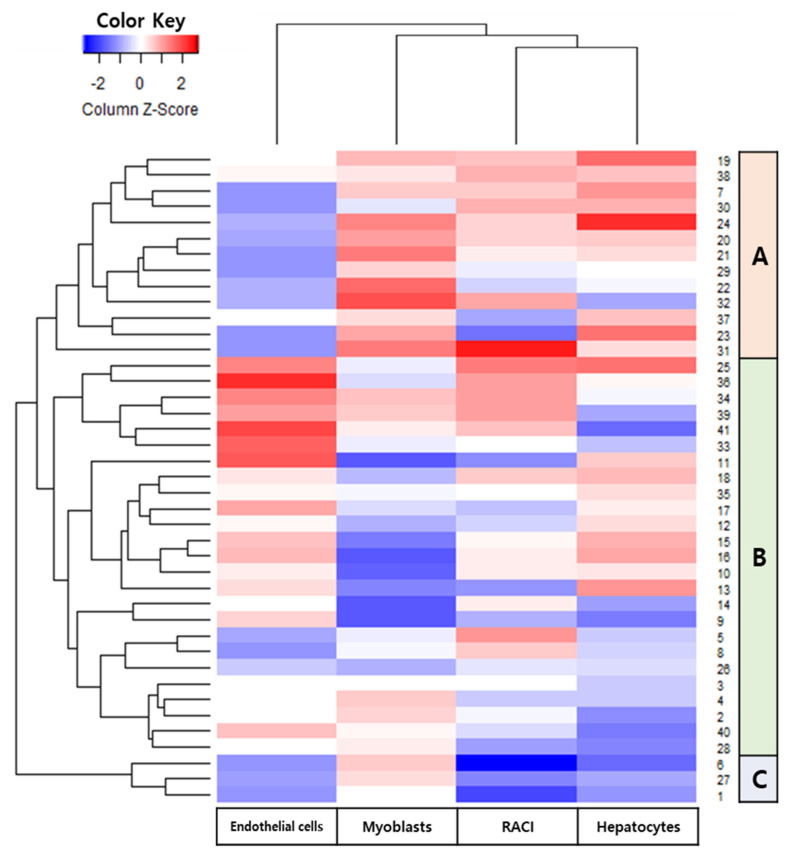
Hierarchical clustering analysis of antioxidant and cytoprotective capacities of whole wheat extracts (WWE).

**Table 1 foods-11-02338-t001:** List of 41 wheat cultivars.

No.	Cultivars	Cross-Combination	No.	Cultivars	Cross-Combination
1	Ol	Norin 72/Norin 12	21	Jonong	SW234/SW80199-B-Y14-0
2	Geuru	F1(Strampelli/69D3607)/Chokwang	22	Jokyung	Seri82/Keumkang
3	Dahong	Norin 72/Weongwang	23	Yeonbaek	Keumkang/Tapdong
4	Chungkye	Norin 4/Sharbatisonora	24	Shinmichal 1	Alchan//Kanto107/Baihuo
5	Eunpa	Chgoku 81/Tob-CNO/Yukseung3/Sw185	25	Dabun	Suwon234//SW26039/Suwon220/3/Keumkang
6	Tapdong	Chugoku 81//SW158/Toropi	26	Baekjoong	Keumkang/Olgeuru
7	Namhae	Olmil/Calidad	27	Jeokjoong	Keumkang/Tapdong
8	Uri	Geuru/Ol	28	Sukang	Suwon266/Asakaje
9	Olgeuru	Geuru’s’/Chokwang//Nishigai143	29	Hanbaek	Shann7859/Keumkang//Guamuehill
10	Alchan	Suwon210/Tapdong	30	Suan	Keumkang/Eunpa//Keumkang
11	Gobun	(Eunpa/Shannung6521)	31	Dajoong	SW90149-B-1-SE3-3-2/Keumkang
12	Keumkang	F1Geuru‘S’/Kanto75//Eunpa	32	Goso	IRENA/Olgeuru
13	Seodun	Geurumil/Genaro 81	33	Joah	SW86054-MB-27-3-2-1-1-1/Sumai#3
14	Saeol	Shirogane//F1(norin 43/Sonalika)	34	Hojoong	Alchan*2/3/Chunm18//JUP/BJY/4/Keumkang
15	Jinpoom	Geurumil/Genaro81	35	Baekchal	Keumkang/Shinmichal
16	Milsung	Sirogane//Norin 43/Sonalika	36	Jojoong	Suwon272/Olgeuru//Keumkang/Suwon252
17	Joeun	Eunpamil/Suwon242	37	Baekkang	Topdong/Klasic
18	Anbaek	Saemil/Geurumil	38	Seaekeumkang	Keumkang/Olgeuru
19	Jopoom	SW88416-B-0/SW89277(F1)	39	Taejoong	XIAN83(104).11/Keumkang
20	Shinmichal	Olgeuru/kwandong 107/Baihuo	40	Johan	96PYT115/Suwon262//Joeun
			41	Hwanggeum	Jokyung/Suwon293

**Table 2 foods-11-02338-t002:** Descriptive statistics of antioxidant properties and cytoprotective activities in whole wheat extracts (WWE).

	Min	Max	Mean	SD ^1^	Skewness	Kurtosis	CV ^2^ (%)
Antioxidant properties							
Total phenolic contents (GAE ^3^ mg/g residue)	9.62	12.23	10.87	0.50	−0.02	0.81	4.62
DPPH radical scavenging (TE ^4^ mg/g residue)	3.17	3.89	3.57	0.19	−0.20	−0.73	5.19
ABTS radical scavenging (TE mg/g residue)	27.73	57.16	45.04	5.34	−0.74	2.02	11.86
Cytoprotective activities ^5^ (%)							
Hepatocytes	9.75	138.29	63.97	33.01	0.11	−0.83	51.61
Myoblasts	0.00	118.97	58.83	32.69	−0.33	−0.54	55.57
Endothelial cells	0.00	118.41	39.89	35.08	0.51	−0.69	87.92

^1^ Standard deviation. ^2^ Coefficient of variation. ^3^ Gallic acid equivalents. ^4^ Trolox equivalents. ^5^ Hepatocytes, cytoprotective activity in hepatocytes (HepG2); Myoblasts, cytoprotective activity in myoblasts (C2C12); Endothelial cells, cytoprotective activity in endothelial cells (EA.hy926).

**Table 3 foods-11-02338-t003:** Pearson’s correlations among antioxidant properties and cytoprotective activities in whole wheat extracts (WWE).

Parameters	RACI	TPC	DPPH	ABTS	Hepatocytes	Myoblasts	Endothelial Cells
RACI	1	0.797 **	0.634 **	0.702 **	0.276	0.165	0.139
TPC		1	0.280	0.419 **	0.104	0.120	0.143
DPPH			1	0.076	0.280	0.024	0.030
ABTS				1	0.208	0.208	0.121
Hepatocytes					1	0.007	−0.071
Myoblasts						1	−0.460 **
Endothelial cells							1

** Pearson’s correlation *p* < 0.01. RACI, relative antioxidant capacity index; TPC, total phenolic content; DPPH, DPPH radical scavenging activity; ABTS, ABTS radical scavenging activity; hepatocytes, cytoprotective activity in hepatocytes (HepG2); myoblasts, cytoprotective activity in myoblasts (C2C12); endothelial cells, cytoprotective activity in endothelial cells (EA.hy926).

**Table 4 foods-11-02338-t004:** Principal component analysis (PCA) of the antioxidant and cytoprotective capacities of whole wheat extracts (WWE); eigenvalues and percentage variability explained by the first four components.

Parameters ^1^	PC1	PC2	PC3	PC4
RACI	0.17	0.72	0.51	−0.44
Hepatocytes	0.21	0.62	−0.70	0.27
Myoblasts	0.70	−0.07	0.38	0.60
Endothelial cells	−0.66	0.31	0.31	0.61
Eigenvalue	1.48	1.28	0.83	0.41
Variability (%)	36.89	32.04	20.72	10.36
Cumulative variability (%)	36.89	68.93	89.64	100.00

^1^ RACI, relative antioxidant capacity index; hepatocytes, cytoprotective activity in hepatocytes (HepG2); myoblasts, cytoprotective activity in myoblasts (C2C12); endothelial cells, cytoprotective activity in endothelial cells (EA.hy926).

**Table 5 foods-11-02338-t005:** Average cluster values of antioxidant and cytoprotective capacities of whole wheat extracts (WWE).

Groups	No. acc.	RACI	Hepatocytes	Myoblasts	Endothelial Cells
A	13	0.24 ^a^	87.29 ^a^	88.90 ^a^	12.36 ^b^
B	25	0.04 ^a^	56.64 ^a^	41.89 ^b^	58.89 ^a^
C	3	−1.42 ^b^	24.03 ^b^	69.63 ^a^	0.94 ^b^

Different letters in (a,b) in the same column indicate significant differences determined using Duncan’s multiple range test (*p* < 0.05). RACI, relative antioxidant capacity index; hepatocytes, cytoprotective activity in hepatocytes (HepG2); myoblasts, cytoprotective activity in myoblasts (C2C12); endothelial cells, cytoprotective activity in endothelial cells (EA.hy926).

## Data Availability

Data is contained within the article or Appendix A.

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
