# Peer review of "Antioxidant and Cytoprotective Capacities of Various Wheat (Triticum aestivum L.) Cultivars in Korea"

_foods, 2022, doi:10.3390/foods11152338_

Round 1

Reviewer 1 Report

I believe this manuscript could be considered for publication. I just have some minor observations.

Author Response

1. In abtract: Units in relative antioxidant capacity index?

Response 1) RACI is in a numerical scale with no units. More explanations are included in response 4.

2. These sentences are repeated, please modify them.

Responses 2) The sentences are modified from “This study provides insights into the variation in the antioxidant and cytoprotective profiles of wheat cultivars. The findings on the antioxidant and cytoprotective capacities of 41 different wheat cultivars provide information for the development of wheat varieties valuable as functional food crops and new dietary ingredients.” to “The results of this study highlight the variation in the antioxidant and cytoprotective capacities of 41 different wheat cultivars. This study provides basic information that can guide decisions in wheat breeding programs for the development of functional food crops and new dietary ingredients.”

3. In M&M, 2.3. Antioxidant capacities and total phenolic content in WWE: Standards used? How did you expressed the results?

Responses 3) Trolox and gallic acid were used as standards for antioxidant capacities and total phenolic content, respectively. We added the following sentence “TPC was expressed as mg gallic acid equivalent and DPPH and ABTS radical scavenging activities were expressed as Trolox equivalent antioxidant capacity.” in line 80-81.

4. Why did you used this score? Not clear.

Response 4) For determining antioxidant activity of foods, the results from different methods may vary substantially because each complex chemical reaction generates unique values. The relative antioxidant capacity index (RACI) can be used to get complete and dynamic picture of the ranking of the test samples. RACI is created from the perspective of statistics by integrating the antioxidant values generated from different in vitro methods. In this study, 41 different wheat cultivars were compared for their antioxidant capacities with different methods. To get a complete ranking of the samples, RACI was used in this study.

Reviewer 2 Report

See attached file

Author Response

Reviewer 2

Review - Manuscript ID: FOODS 1826480

“Antioxidant and Cytoprotective Capacities of Various Wheat (Triticum aestivum L.) Cultivars”

General and major remarks

The work presented addresses an interesting subject. However, this study, which aims to study antioxidant and cytoprotective capacities of various wheat (Triticum aestivum L.) cultivars, deserves to be thoroughly reviewed. Indeed, the work presented has several serious shortcomings which are extensively detailed later in the comments.

Major remarks

Lines 45-46: If the reader refers to the words of the authors, “the objective of this study was to determine the antioxidant and cytoprotective capacities of various wheat cultivars”. A list of wheat genotypes was retained (Table 1) to carry out this study. However, the reasons for this choice are never made explicit. Thanks to the authors for providing the reasons for the choice of plant material. At the same time, the conditions for obtaining this plant material are not specified. However, as the authors themselves point out (lines 125-127), all growth conditions affect phenolic compounds in wheat cultivars. Thus, how can we compare different genotypes if they were not grown under the same agronomic and/or physiological conditions?

In the absence of precise information on the plant material used, it is therefore very difficult for the reader to assess (i.e. genotypic variance versus environmental variance for example) the results presented (Table 2).

Response) In Korea, various wheat cultivars are developing in Rural Development Administration (RDA). Up to now about 45 cultivars are developed in Korea and among them, 41 different cultivars are evaluated in this study. The samples were provided by RDA. The 41 different wheat cultivars were grown in same area and same period. More detailed information was given in “M&M, 2.1. Plant materials”. Also, the reason for choosing these 41 samples was the cultivars developed in Korea. Therefore, we added ‘in Korea’ in title and ‘breeded in Korea’ in line 53.

Lines 46-47: Still according to the words of the authors, “this study also aimed to analyze the correlation between antioxidant and cytoprotective activities”. On reading the current version of the results presented in paragraphs 3.2, 3.3 and 3.4, the reader really has a hard time following the intellectual approach followed by the authors. Thus:

  • Thus, after completely reviewing the data in Table 3 based on a new analysis of the data (Table S1), it seems that the statistical significance of several values is erroneous (see table below).

If we consider this new information, it clearly appears that little or even very little relationship exists between the antioxidant properties (TPC, DPPH and ABTS) measured and the cytoprotective activities observed, contrary to what is asserted by the authors in paragraph 3.2.

  • In the same way, if we observe the results presented by the authors in table 5 at the end of a hierarchical clustering analysis, it seems relatively obvious that despite the fact that 3 groups have been defined (exclusively from cytoprotective activities), these do not differ statistically between them on the basis of antioxidant properties (same letter for all using Duncan's multiple range test). This therefore seems to indicate that the variables retained to qualify the antioxidant properties are not relevant to explain the natural variability of the cytoprotective capacities.
  • Finally, it is very difficult to understand the implementation here of a principal component analysis of the data which are relatively explicit on their own. Thank you to the authors for justifying this choice and the contribution of this one in the explanatory process?

Response) We totally agree with you and whole manuscript is revised.

First, Pearson’s correlations (Table 3) were corrected. We were used three replicates for Table 3 and now we used the means of three replicates to show Table 3. In addition, RACI was added in Table 3. We concluded that RACI was significantly related with antioxidant parameters and no correlation was found between RACI and cytoprotective capacities. Therefore, we added possible reason for the no correlation. Following sentences are included in line 164-777. “Significant correlations were observed between the antioxidant properties of RACI-TPC (P < 0.01), RACI-DPPH (P < 0.01) and RACI-ABTS (P < 0.01) meaning RACI is a reasonably accurate rank of antioxidant capacity among wheat cultivars. However, no significant correlations between RACI and cytoprotective activities were ascertained. This might be due to the presence of other bioactive compounds including alkylresorcinols, phytosterols, and policosanols in wheat. Phytosterols are structurally related to cholesterol and reduce serum low-density lipoprotein cholesterol levels [17]. According to a previous report β-sitosterol inhibited muscle atrophy in muscle atrophy C2C12 cells [18]. Alkylresorcinols exhibit antifungal and antibacterial activities [19,10]. Octacosanol enhances the migration and proliferation of human umbilical vein endothelial cells [21]. However, De Jong et al. (2008) found that phytosterol consumption did not affect antioxidative enzyme activity, endothelial dysfunction, and low-grade inflammation [22]. Therefore, further research is necessary to study the relationship between cytoprotective activity and phytochemical content.”

Second, PCA analysis (Table 4), 2D scatter diagram of principal component analysis (Figure 2), Hierarchical clustering analysis (Figure 3), and Average cluster values (Table 5) were changed to include RACI because RACI was added in Table 3.

For PCA, following sentences are included in line 202-214. “PCA was used to detect clustering and to establish relationships between cultivars and antioxidant and cytoprotective capacities in this study. The data were subjected to PCA (Table 4). The eigenvalues assist in defining four factors (RACI, hepatocytes, myoblasts, and endothelial cells) that can be retained. The sum of eigenvalues is generally equivalent to the number of variables [23]. Of the four principal components (PCs), PC1 and PC2 had eigenvalues > 1 and could explain 68.93% of the total cumulative variability. The contribution of PC1 to variability was the highest (36.89%). Numerals with the highest absolute value nearer to unity in the first PC affect the grouping more than those with a lesser absolute value nearer to zero [24]. Hence, activity in myoblasts had the largest positive loading in PC1. The first two PCs, which contributed 68.93% of the total variance, were plotted as biplots on PC1 and PC2 on the X- and Y-axes, respectively, to detect the association between clusters (Figure 2). RACI and activity in hepatocytes were the highest relationship in 41 cultivars.”

For clustering analysis, following sentences are included in line 245-249. “Group A contained 13 cultivars and had the highest RACI and cytoprotective activities in hepatocytes and myoblast. Group B contained 25 cultivars and had the highest cytoprotective activities in the endothelial cells. Group C contained 3 cultivars and had relatively lowest antioxidant and cytoprotective activities in the hepatocytes and endothelial cells.”

The contents of the summary and the conclusion will have to be thoroughly revised following a general revision of the current version of the work.

Response) The abstract and conclusions were changed according to the changes in main text. Please refer the revised version.

Reviewer 3 Report

Dear author

I report a review result about ‘Antioxidant and Cytoprotective Capacities of Various Wheat (Triticum aestivum L.) Cultivars’. There are some revision points. Please confirm the item which I pointed out. Please inform it if there are my deficiency.

Best regard.

1. Cultivation condition of provided 41cultivars

Author described 41 cultivars were provided by the Rural Development Administration of the Republic of Korea in 2019. Were these cultivars cultivated on the same condition? Author revealed that antioxidant is affected by not only variety but also environment (L124-128). I agree, too. If the condition of these samples is not clear, it is not the characteristics of cultivars but the mere introduction of analysis data.

2. What is whole wheat extract (WWE)?

What extract is this? I know it if I read front and back it pointed the whole grain, but explanation is insufficient. Recently, the extract of the leaf is attracted attention. Please describe that WWE is whole wheat grain extract in the beginning of this article.

3. 1000 grain weight

Does author have the data of 1000 grain weight of 41 cultivars? It seems that small grain tends to show high antioxidant. If had these data, should discuss the relationship among them.

4. Cytoprotective differences of hepatocytes, myoblasts, and endothelial cells.

Why are effects different for these cells? Is it the characteristic of the organ? It may be different in activity of the cell division. Please describe it as far as you know.

5. Advice for breeding plan

I think that hierarchical clustering analysis is useful for a classification of cultivars. Should breeders crossbreed it with a different group whether they should crossbreed it in the same group? Please advise for breeders.

Author Response

Dear author

I report a review result about ‘Antioxidant and Cytoprotective Capacities of Various Wheat (Triticum aestivum L.) Cultivars’. There are some revision points. Please confirm the item which I pointed out. Please inform it if there are my deficiency.

Best regard.

1. Cultivation condition of provided 41cultivars

Author described 41 cultivars were provided by the Rural Development Administration of the Republic of Korea in 2019. Were these cultivars cultivated on the same condition? Author revealed that antioxidant is affected by not only variety but also environment (L124-128). I agree, too. If the condition of these samples is not clear, it is not the characteristics of cultivars but the mere introduction of analysis data.

Response) We added detailed information on cultivation in “M&M, 2.1. Plant materials”. Following sentences are added. “Whole wheat cultivars were provided by the Rural Development Administration of the Republic of Korea in 2019. Korean wheat cultivars were sown in randomized complete blocks with 3 replicated in the upland crop experimental farm of National Institute of Crop Science (NICS) of the Rural Development Administration (RDA, Republic of Korea). The seeds were sown on October 25, 2018 and plots were combine-harvested on June 16, 2019. Fertilizer was applied at 9.1: 7.4: 3.9 kg/10a (N: P: K) before sowing. Weeds, insects and disease were stringently controlled. No supplemental irrigation was applied. Grain from each plot was dried using forced air driers and bulked from replications to provide grain for quality analysis. A list of 41 different wheat cultivars and their cross-combination information is shown in Table 1.”

2. What is whole wheat extract (WWE)?

What extract is this? I know it if I read front and back it pointed the whole grain, but explanation is insufficient. Recently, the extract of the leaf is attracted attention. Please describe that WWE is whole wheat grain extract in the beginning of this article.

Response) In abstract, ‘whole wheat extracts (WWE)’ was changed to ‘whole wheat grain extracts (WWE)’. Also, the title and sentence in 2.2 (lines 70 and 71) were changed to have grain.

3. 1000 grain weight

Does author have the data of 1000 grain weight of 41 cultivars? It seems that small grain tends to show high antioxidant. If had these data, should discuss the relationship among them.

Response) In this study, about 30 to 50 grams of each cultivar was weighed and milled. This weight is equivalent to about 700-1,200 grains depending on cultivars. We think that this weight or grains are homogeneous.

4. Cytoprotective differences of hepatocytes, myoblasts, and endothelial cells.

Why are effects different for these cells? Is it the characteristic of the organ? It may be different in activity of the cell division. Please describe it as far as you know.

Response) This is a natural phenomenon. It is not each to elucidate why different molecules or extracts act differently in different cells. Therefore, we mentioned “Therefore, further research is necessary to study the relationship between cytoprotective activity and phytochemical content.” in line 175-177.

5. Advice for breeding plan

I think that hierarchical clustering analysis is useful for a classification of cultivars. Should breeders crossbreed it with a different group whether they should crossbreed it in the same group? Please advise for breeders.

Response) We added following sentences in line 245-249. “Group A contained 13 cultivars and had the highest RACI and cytoprotective activities in hepatocytes and myoblast. Group B contained 25 cultivars and had the highest cytoprotective activities in the endothelial cells. Group C contained 3 cultivars and had relatively lowest antioxidant and cytoprotective activities in the hepatocytes and endothelial cells.”

Round 2

Reviewer 2 Report

Review - Manuscript ID: FOODS 1826480 Step2

Antioxidant and Cytoprotective Capacities of Various Wheat (Triticum aestivum L.) Cultivars”

General remarks

After a complete and careful proofreading of the new version proposed by the authors, I can only confirm the significant improvements introduced in each of the sections of the text. However, some elements deserve further correction:

Thus, in the context of paragraph 3.2., the authors comment on the statistical relationships which exist between the "RACI" variable and the 3 other variables characterizing the antioxidant properties.

It is very surprising to comment on these elements insofar as the "RACI" variable is a variable constructed from a linear combination of the 3 others (therefore necessarily not independent of the 3 initial variables)! I therefore see absolutely no point in these elements of discussion.

The same remark can be made for the abstract and the first paragraph of the conclusion.

Lines 279-280: The authors repeat the sentence “The correlation between RACI and cytoprotective activities was weak”. Strictly speaking, I think these should use the same sentence proposed in the summary (lines 20-21).

Again, it is very difficult to understand the implementation here of a principal component analysis of the data. As requested previously, thank you to the authors for justifying this choice and the contribution of this one in the explanatory process?

Minor remarks

Table 2: For the sake of homogeneity, we can suggest that the authors add elements of descriptive statistics concerning the "RACI" variable in this table.

Author Response

General remarks

After a complete and careful proofreading of the new version proposed by the authors, I can only confirm the significant improvements introduced in each of the sections of the text. However, some elements deserve further correction:

Response) We are grateful to you for your valuable time and comments. We revised the manuscript with reference to the suggestions.

1. Thus, in the context of paragraph 3.2., the authors comment on the statistical relationships which exist between the "RACI" variable and the 3 other variables characterizing the antioxidant properties.

It is very surprising to comment on these elements insofar as the "RACI" variable is a variable constructed from a linear combination of the 3 others (therefore necessarily not independent of the 3 initial variables)! I therefore see absolutely no point in these elements of discussion.

The same remark can be made for the abstract and the first paragraph of the conclusion.

Response) Comments on RACI and 3 other variables characterizing the antioxidant properties were deleted in abstract, paragraph 3.2., and conclusion according to your suggestion. Please refer line 18-19 in abstract, line 160-163 in paragraph 3.2., and line 246-247 in conclusions.

2. Lines 279-280: The authors repeat the sentence “The correlation between RACI and cytoprotective activities was weak”. Strictly speaking, I think these should use the same sentence proposed in the summary (lines 20-21).

Response) The sentence “The correlation between RACI and cytoprotective activities was weak. ” was changed with “However, no significant correlations between antioxidant and cytoprotective capacities were ascertained.” Please refer line 247-252.

3. Again, it is very difficult to understand the implementation here of a principal component analysis of the data. As requested previously, thank you to the authors for justifying this choice and the contribution of this one in the explanatory process?

Response) PCA was used to define the discriminant effects of RACI and cytoprotective activities between the wheat cultivars. HCA was used to classify wheat cultivars. We revised PCA paragraphs with more information and explanations. Please refer line 181-210. In addition, we revised some sentences in conclusions according to the changes in PCA paragraphs. Please refer line 252-255.

Minor remarks

4. Table 2: For the sake of homogeneity, we can suggest that the authors add elements of descriptive statistics concerning the "RACI" variable in this table.

Response) We tried to add RACI in Table 2. However, as you know, the RACI was calculated values from three antioxidant parameters. The RACI has almost zero mean values and, therefore, it has extremely high CV. Therefore, we did not add RACI in Table 2.